# RNAi-Based Therapy: Combating Shrimp Viral Diseases

**DOI:** 10.3390/v15102050

**Published:** 2023-10-05

**Authors:** Md. Shahanoor Alam, Mohammad Nazrul Islam, Mousumi Das, Sk. Farzana Islam, Md. Golam Rabbane, Ehsanul Karim, Animesh Roy, Mohammad Shafiqul Alam, Raju Ahmed, Abu Syed Md. Kibria

**Affiliations:** 1Department of Genetics and Fish Breeding, Bangabandhu Sheikh Mujibur Rahman Agricultural University, Gazipur 1706, Bangladesh; msalambd@bsmrau.edu.bd; 2Department of Biotechnology, Sher-e-Bangla Agricultural University, Dhaka 1207, Bangladesh; nazrul.sau.bd@gmail.com; 3Department of Aquaculture, Bangabandhu Sheikh Mujibur Rahman Agricultural University, Gazipur 1706, Bangladesh; mousumi.aqc@bsmrau.edu.bd; 4Department of Fisheries (DoF), Government of the People’s Republic of Bangladesh, Matshya Bhaban, Ramna, Dhaka 1000, Bangladesh; skfarzanaislam@yahoo.com (S.F.I.); rajuahmeddof@gmail.com (R.A.); 5Department of Fisheries, Faculty of Biological Sciences, University of Dhaka, Dhaka 1000, Bangladesh; rabbane@du.ac.bd; 6Bangladesh Fisheries Research Institute, Mymensingh 2201, Bangladesh; ehsan_tony@yahoo.com; 7Department of Fisheries Biology and Aquatic Environment, Bangabandhu Sheikh Mujibur Rahman Agricultural University, Gazipur 1706, Bangladesh; animesh10@bsmrau.edu.bd; 8Department of Aquaculture, Hajee Mohammad Danesh Science and Technology University, Dinajpur 5200, Bangladesh; kibriahstu@gmail.com

**Keywords:** penaeid shrimp, shrimp viral diseases, innate immune response, dsRNA, siRNA, therapeutics, post-transcriptional gene silencing

## Abstract

Shrimp aquaculture has become a vital industry, meeting the growing global demand for seafood. Shrimp viral diseases have posed significant challenges to the aquaculture industry, causing major economic losses worldwide. Conventional treatment methods have proven to be ineffective in controlling these diseases. However, recent advances in RNA interference (RNAi) technology have opened new possibilities for combating shrimp viral diseases. This cutting-edge technology uses cellular machinery to silence specific viral genes, preventing viral replication and spread. Numerous studies have shown the effectiveness of RNAi-based therapies in various model organisms, paving the way for their use in shrimp health. By precisely targeting viral pathogens, RNAi has the potential to provide a sustainable and environmentally friendly solution to combat viral diseases in shrimp aquaculture. This review paper provides an overview of RNAi-based therapy and its potential as a game-changer for shrimp viral diseases. We discuss the principles of RNAi, its application in combating viral infections, and the current progress made in RNAi-based therapy for shrimp viral diseases. We also address the challenges and prospects of this innovative approach.

## 1. Introduction

Modern shrimp farming commenced in Japan in 1933 when they successfully induced artificial spawning and hatching larvae of *Marsupenaeus japonicus* (Spence Bate) [1]. Subsequently, shrimp aquaculture evolved into a commercial enterprise in Asia during the 1950s [2], and in America during the 1970s [3]. In recent decades, this sector has experienced rapid expansion owing to rising consumer demand, leading to substantial contributions to the socio-economic progress of coastal communities in numerous developing nations [4]. As this industry generates significant levels of income in underdeveloped countries, it aids in reducing economic gaps between nations [5]. In fact, penaeid shrimp have made a considerable economic contribution to Southeast Asia’s countries of Thailand, Vietnam, Indonesia, and the Philippines [6].

Stress factors have enhanced disease susceptibility as a result of rapid growth and intensification of shrimp farming [7,8]. However, the expansion of worldwide shrimp aquaculture production has encountered a significant hurdle due to the rise in frequent disease outbreaks, resulting in massive mortality within the industry [9]. As a result, shrimp diseases pose significant challenges to the aquaculture industry, leading to substantial economic losses worldwide. Infectious agents such as bacteria, fungi, and viruses, have emerged as major culprits responsible for devastating shrimp populations [10,11]. The majority of shrimp diseases result from viral infections, and their detrimental impacts are almost four times stronger compared to bacterial diseases [9].

Most of the time, appropriate shrimp farm management practices can stop parasitic and bacterial diseases from spreading, but this preventive approach may not be as effective against viral diseases [12,13]. Conventional approaches to disease management, including antibiotics and vaccines, have limitations in terms of efficacy, specificity, and environmental impact [14,15,16]. As a result, there is a pressing need for innovative therapeutic strategies that can effectively combat these diseases while minimizing their detrimental effects. Most recently, RNA interference (RNAi) technology has gained considerable attention as a promising tool for the treatment of shrimp diseases. RNAi is a conserved cellular mechanism that regulates gene expression by suppressing the activity of specific genes [17,18]. It offers a targeted approach to interfere with the replication and pathogenesis of infectious agents, making it an attractive alternative for disease control in shrimp aquaculture. This review provides an overview of the present situation and upcoming approaches related to the therapeutic use of RNAi to combat viral diseases in shrimp.

## 2. Viral Diseases in Shrimp

Viral infection causes mortality, slow growth, and deformities in shrimp at different ages of their life. Shrimps are susceptible to a variety of viral diseases, and the viral pathogens can spread rapidly within aquaculture systems. They are vulnerable to over 20 different viruses, belonging to more than 10 distinct families [17] (Table 1). Among these, white spot syndrome virus (WSSV), yellow head virus (YHV), Taura syndrome virus (TSV) and infectious hypodermal and hematopoietic necrosis virus (IHHNV) have been extensively studied and their characteristics well understood [4,19]. These are particularly dangerous because of their widespread occurrence and significant impact on the economy.

## 3. The Antiviral Immunity in Shrimp

Viral pathogens have developed strategies to evade shrimp’s immune defenses. Shrimp, in contrast to vertebrates, only have innate immunity, a type of non-specific immune response made up mostly of cellular defense and humoral defense [32]. Cellular immunity is a vital component of the immune response in organisms, including shrimp. It encompasses various processes, such as phagocytosis, killing, and elimination of pathogens, as well as apoptosis, to combat microbial infections [32,33]. Shrimps also produce antimicrobial peptides that can directly target and destroy viruses [34]. Additionally, they can generate reactive oxygen species (ROS), which are toxic molecules that can damage viral particles and infected cells [35]. This cellular immune response plays a crucial role in recognizing and removing pathogens, thereby maintaining the overall health and integrity of the organism.

In recent years, an increasing number of investigations have reported evidence highlighting the significance of RNA interference (RNAi) as a critical strategy for shrimp in defending against viral invasions. Initially identified in plants and nematodes, subsequent studies revealed its presence in various eukaryotes [36]. Increasing evidence suggests that RNAi serves as a crucial antiviral immune response in animals. 

## 4. Current Therapeutic Strategies to Minimize the Impacts of Viral Diseases

In the past 20 years, researchers devised and tested numerous strategies in experimental settings to combat the detrimental effects of viral diseases, specifically focusing on WSSV in shrimp aquaculture. Products that have been evaluated include:

### 4.1. Different Antiviral Agents

In the past decade and a half, several natural antiviral agents have been examined for their efficacy in shrimp against WSSV. These antiviral substances were delivered orally to shrimp before a WSSV challenge. The outcomes of a research investigation that involved a diet enriched with a derivative from Spirulina demonstrated no noticeable antiviral benefits, but instead, a slight delay in the rate of mortality [37]. In contrast, the oral application of a compound called bis[2-methylheptyl]phthalate derived from the leaves of *Pongamia pinnata* effectively hindered the WSSV progression and reduced the mortality rate among affected shrimp (ranging from 60% to 20%) [38]. In the sole study on using the synthetic antiviral agent cidofovir against WSSV infection, it outperformed the Spirulina-supplemented diet in reducing and delaying mortality during intramuscular WSSV challenge. However, cidofovir was unable to hinder the occurrence of WSSV infection [37].

### 4.2. Immunostimulants

These products are derived from various sources including algae (*Sargassum polycystum*) [39], herbs [40], fungi (*Saccharomyces cerevisiae*, *Schizophyllum commune*) [41,42], and bacteria (*Bacillus* sp.) [41]. These life forms have cellular walls comprising elements such as peptidoglycans, β-glucans, and/or lipopolysaccharides. These compounds trigger both humoral and cellular immune responses in shrimp [41,43]. Experimental animals are fed immunostimulants prior to and at the time of the WSSV challenge, and the outcomes indicate a decrease in mortality when compared to untreated groups [39,40,44,45]. However, it is important to note that prolonged utilization of immunostimulants could result in immune system exhaustion in shrimp [45,46]. This can result in decreased effectiveness of the immunostimulants and potentially have adverse effects on the shrimp’s immune system.

### 4.3. DNA Vaccines

A vaccine is a form of biological preparation designed to enhance immunity against a particular disease or a set of diseases [47]. Initially, the method of electroporation was used to introduce foreign DNA into *P. monodon* eggs and embryos in shrimp. The effectiveness of this approach varied between 37% and 19%, and the rate at which transgenic eggs progressed into juvenile shrimp with viability was merely 0.6% [48]. In another study conducted on *M. japonicus*, DNA vaccine was delivered to embryos through particle bombardment, electroporation, and microinjection. Out of these methods, microinjection was identified as the most successful, as it facilitated the delivery of greater quantities of foreign DNA [49]. Recent studies evaluated the protective efficacy of DNA vaccines against WSSV by encoding WSSV envelope proteins such as vp281, vp35, vp28, and vp15 [50,51]. While DNA vaccines have been successful in some contexts, their efficacy in combating shrimp viral diseases appears limited. The intricate immune response of shrimp and the complexities of their viral infections may pose hurdles for DNA vaccines to induce adequate protection. They may not be the most suitable solution for addressing viral diseases in shrimp.

### 4.4. Changing the Water Temperature

Studies have shown that manipulating the temperature of the aquatic environment can significantly reduce the incidence and severity of viral infections among shrimp populations. Raising the water temperature to 32 °C led to a decrease in virus reproduction and a reduction in shrimp mortality (ranging from 0% to 30%), in contrast to the 100% mortality observed in specimens kept at 27 °C [37,52]. The method of WSSV inoculation did not affect the protective effects of high-temperature treatment [53]. Hyperthermia still had a significant beneficial effect even when maintained at 33 °C for 18 h, with mortality ranging from 0% to 40% [52]. Studies carried out on both shrimp and crayfish have revealed that while hyperthermia reduces virus replication, the organisms still retain a certain level of infection, which is detectable through real-time PCR methods [54,55]. Similarly, reduced water temperature proves to be efficacious in suppressing virus replication among species inhabiting temperate or cold-water regions. For *M. japonicus* shrimp, a temperature of 15 °C exhibited better suppression of WSSV replication compared to 33 °C [56]. Similarly, crayfish species such as *Astacus astacus* (Linnaeus), *Procambarus clarkii* (Girard), and *Pacifastacus leniusculus* (Dana) kept at temperatures of 4, 10, or 12 °C, exhibited no mortality when subjected to WSSV infection. Conversely, animals infected with WSSV and maintained at temperatures between 22 °C and 24 °C experienced complete mortality [57,58]. Even though the exact process causing the suppression of virus replication is not clear, there is a proposition that hyperthermia could trigger apoptosis in infected cells [55] thereby halting viral progression. An alternative theory suggests that hyperthermia might disturb the crucial biochemical characteristics of enzymes required for virus replication. This disruption could impede replication even as the animals still remain infected [37]. Hyperthermia, while considered a potential strategy to impede viral replication in shrimp, may not guarantee a complete halt in the replication process. Despite its inhibitory effects, viral replication might persist to some extent even under elevated temperatures.

### 4.5. RNAi Based Therapy

Recently, RNAi technology has been studied in the context of shrimp farming as a way to target and reduce the expression of genes linked to shrimp diseases. RNAi is a natural process that occurs after transcription, where double-stranded RNA (dsRNA) induces the degradation of mRNA transcripts that are homologous to it [15,59]. The multidomain ribonuclease III enzyme dicer cleaves the dsRNA into fragments of 21 to 23 nucleotides (nt), with characteristic 2-nt 3’ overhangs [60,61,62]. These specific dsRNA fragments, known as small interfering RNAs (siRNAs), confer sequence specificity in subsequent steps that lead to the degradation of target viral mRNA inhibiting viral replication [63]. In shrimp, the use of vp28 and vp37 siRNA effectively eliminates WSSV virions from *M. japonicus* [64,65] and *Litopenaeus vannamei* (Boone) [66,67] shrimp, challenged with WSSV, demonstrating the efficiency of siRNA-based RNAi as a technique to combat viral infections in shrimp. YHV-specific dsRNA-injected shrimp targeting the *RdRp* and *rr2* genes do not exhibit yellowhead disease after the YHV challenge [68,69,70]. Numerous RNAi-linked genes in shrimp have been identified across different species of shrimp (Table 2).

## 5. Basic Mechanism of RNAi

In 2005, multiple research teams reported that the administration of long dsRNA molecules designed for specific viral genes substantially protected shrimp from a fatal viral infection by silencing the activity of the targeted viral genes [71,72,73]. Basically, RNAi is a widely conserved mechanism that regulates gene expression by efficiently degrading target messenger RNAs (mRNAs), leading to the silencing of target genes [18]. This process starts when dsRNAs are introduced or produced within a cell [17,62,74]. Upon entering the cytoplasm, these dsRNAs become substrates for an enzyme called Dicer, which has multiple domains and acts as a ribonuclease III [60]. Once Dicer cleaves the dsRNAs into shorter fragments called siRNAs, these siRNAs are recognized by the RNA-induced silencing complex (RISC) [75]. RISC, composed of several enzymes, binds to the double-stranded siRNAs and unwinds them. As a result, the sense strand of the siRNA is released. In certain organisms, this sense strand can trigger the synthesis of additional dsRNA through an enzyme called RNA-dependent RNA polymerase (RdRp). Meanwhile, the antisense siRNA strand remains bound to RISC, serving as a sequence that guides the targeting of the enzyme complex. Once RISC binds to a complementary mRNA molecule, it exhibits its nuclease activity and cleaves the mRNA strand that corresponds to the siRNA sequence. Subsequently, the damaged mRNA is degraded by the cellular machinery, resulting in the specific silencing of the target gene at the post-transcriptional level [17,74].

## 6. Shrimp RNAi as a Virus-Fighting Weapon

Controlling viral diseases in the shrimp industry is still a significant and ongoing concern. Outbreaks of these diseases, particularly in economically important species such as *P. monodon* and *L. vannamei*, have resulted in substantial production losses in various locations since the early 1990s [76]. Shrimp are vulnerable to a variety of pathogens, including bacteria, viruses, fungi, parasites, protozoa, and rickettsia [77]. While bacterial, fungal, and protozoan infections can be controlled with better farming techniques, regular sanitation, and the use of chemotherapeutics, viral pathogens pose a more significant challenge [78].

The precise molecular mechanisms underlying the immune responses against viruses are not yet known for most crustacean species [79]. Therefore, it is crucial to comprehend how viruses enter and spread in shrimp, and the interactions between the host and the pathogen at molecular and cellular levels, to develop effective strategies for combating viral infections in shrimp [80]. The discovery of RNAi has led to the identification of key proteins involved in the RNAi pathway, such as Dicer and Argonaute (Ago), in species such as *P. monodon* [81,82,83], *L. vannamei* [84,85], and *M. japonicus* [86]. This confirms the existence of the RNAi machinery in shrimp. Consequently, specific sequences of dsRNA derived from genes of economically significant shrimp pathogens such as IHHNV, TSV, WSSV, and YHV have been targeted, and their effectiveness in promoting shrimp survival and interfering with viral replication has been demonstrated (Table 2).

Multiple studies have shown that administering pathogen-specific dsRNA/siRNA either before or at the same time as a viral challenge can effectively inhibit the replication of various virus species [64,72,87,88,89]. Conversely, it was also demonstrated that delivering RNAi-inducers to shrimp that are already infected can have a therapeutic effect [90,91,92,93,94]. According to existing literature, silencing of viral structural and non-structural proteins simultaneously [94], as well as targeting both host and viral genes [91], significantly improves the therapeutic response. The potential of RNAi to provide a therapeutic effect in infected shrimp is particularly valuable in hatcheries, as it helps prevent the loss of valuable broodstock due to viral outbreaks.

**Table 2 viruses-15-02050-t002:** Application of RNAi as a defense mechanism against the different viral genes in shrimp aquaculture.

Virus	Target Gene	Host	Delivery Method	RNAi Inducer	Reference
WSSV	*WSSV051*	*P. monodon*	Oral	Bacterial expressed dsRNA	[95]
	*Rab7*	*P. monodon*	Injection	Transcribed dsRNA	[80]
	*Vp28*	*L. vannamei*	Oral	Synthesized	[67]
	*Vp28*	*L. vannamei*	Injection	Transcribed dsRNA	[66]
		*M. ja ponicus*	Injection	Synthesized	[64]
	*ß-integrin*	*M. ja ponicus*	Injection	In vitro transcribed dsRNA	[96]
	*V* *p37*	*L. vannamei*	Injection	Synthesized	[65]
	*rr2*	*L. vannamei*	Injection	Bacterial expressed dsRNA	[70]
	*V9*	*P. monodon*, *M. ja ponicus*	Injection	Synthesized	[97]
	*V26*	*L. vannamei*	Injection	Transcribed dsRNA	[98]
YHV	*Rab7*	*P. monodon*	Injection	Transcribed dsRNA	[80]
	*gp116*, *gp64*	*P. monodon*	Transfection	Transcribed dsRNA	[72]
	*RdRp*	*P. monodon*	Transfection	Transcribed dsRNA	[99]
	*RdRp*	*L. vannamei*	Injection	Bacterial expressed dsRNA	[68]
	*RdRp*	*L. vannamei*	Oral	Microalgal expressed dsRNA	[69]
	*rr2*	*L. vannamei*	Injection	Bacterial expressed dsRNA	[70]
	*YHV-pro*	*P. monodon*	Injection	Bacterial expressed dsRNA	[73]
	*EEA 1*	*P. monodon*	Injection	Bacterial expressed dsRNA	[100]
TSV	*Rab7*	*L. vannamei*	Injection	Bacterial expressed dsRNA	[101]
	*Lamr*	*L. vannamei*	Injection	In vitro transcribed dsRNA	[102]
LSNV	*RdRp*	*P. monodon*	Oral	Bacterial expressed dsRNA	[103]
GAV	*ß-actin*	*P. monodon*	Oral	Bacterial expressed dsRNA	[88]
IMNV	*ORF1a, ORF1b*	*L. vannamei*	Injection	Synthesized	[104]

## 7. siRNA Mediated RNAi

In the process of RNAi mediated by siRNA, exogenous dsRNAs encompassing repetitive sequences and transcripts capable of forming long hairpin structures, undergo processing by Dicer enzyme to generate siRNA duplexes [105]. These siRNA duplexes possess specific characteristics, such as a 3′OH, a 5′ phosphate (PO4), and 3′ dinucleotide overhangs [17]. The siRNA duplex then associates with the Ago protein to form the precursor RNAi-induced silencing complex (pre-RISC). Within this complex, Ago cleaves one strand of the duplex, known as the passenger strand. Eventually, the mature RISC, consisting of an Ago protein and the guide strand, targets complementary mRNA molecules, resulting in translational repression [105] (Figure 1).

In *Penaeus japonicus* (Spence Bate) shrimp, siRNA-mediated RNAi is an essential immune protection mechanism that plays a significant role in responding to viral infections by degrading viral mRNA molecules [64]. This antiviral strategy is based on the premise that viruses produce dsRNA as part of their life cycle [106]. When a shrimp is infected with an RNA virus, the host cells facilitate the replication of viral genomic RNA, leading to the formation of dsRNA precursor molecules. Subsequently, the Dicer2 enzyme processes the dsRNA to generate virus-derived siRNAs, which are then incorporated into a ribonucleoprotein complex known as RISC, containing Ago2 protein [17]. These siRNAs guide the RISC to specifically target and degrade viral mRNAs, thereby inhibiting virus replication [107]. Studies have revealed that siRNAs consist of a seed region (2nd–7th nucleotides) and a supplementary region (12th–17th nucleotides) [108]. The seed region is responsible for the initial recognition of the target, while the supplementary region aids in binding to the target [17]. In the case of *P. monodon* shrimp, it has been observed that in vitro transcribed dsRNAs corresponding to YHV helicase, polymerase, protease, gp116, and gp64 can effectively inhibit the replication of YHV in cultured cells [72]. Moreover, when dsRNA targeting the protease gene of YHV is injected into *P. monodon*, the progression of YHV is notably hindered, leading to a decrease in shrimp mortality [73]. These results highlight the significance of siRNA-mediated RNAi in enhancing the immune response of shrimp against RNA virus invasion [17].

During DNA virus infection, the mRNA transcripts originating from the viral genome have the capacity to shape structures exhibiting double-stranded attributes, referred to as mRNA hairpins. Dicer2 enzyme can identify and process these hairpin structures [109]. The resulting siRNA duplexes are incorporated into the RISC, which facilitates the degradation of viral mRNA molecules, leading to the suppression of virus infection [110]. In *M. japonicus* shrimp, it has been observed that the shrimp can produce a specific antiviral siRNA called vp28-siRNA in response to WSSV infection. In the process of RNAi, against DNA viruses, the essential involvement of shrimp Dicer2 and Ago2 proteins is fundamental for the formation and functioning of vp28-siRNA [109]. These findings highlight the effectiveness of siRNA-mediated RNAi as a strategy employed by animals to combat DNA virus infections [17].

## 8. miRNA Mediated RNAi

miRNAs play a crucial role in regulating numerous biological functions within organisms, including cell development, proliferation, differentiation, metabolism, apoptosis, and immunity [111]. One of the distinctive characteristics of miRNA-mediated regulation is its network-based nature. This means that a single miRNA can target multiple genes simultaneously, and multiple miRNAs can also target the same gene [112,113,114,115,116]. Such complex interactions between miRNAs and their target genes contribute to the intricate regulatory networks that govern diverse biological processes.

In the pathway of RNAi mediated by miRNA, miRNAs are produced from noncoding RNA transcripts or small introns that fold into partially formed stem-loop structures [117]. The process begins with the transcription of primary miRNAs (pri-miRNAs) by RNA polymerase. These pri-miRNAs are then trimmed by an enzyme called Drosha, resulting in the production of ~70 nucleotide precursors called pre-miRNAs (Figure 1). The pre-miRNAs, bound to exportin 5, are transported from the nucleus to the cytoplasm. In the cytoplasm, the pre-miRNAs are further processed by an enzyme called Dicer, resulting in the formation of miRNA duplexes. The miRNA duplexes associate with Ago proteins to form a precursor RNA-induced silencing complex (pre-RISC). The passenger strand of the duplex is subsequently removed, leaving behind the mature RISC complex that contains the guide strand. This mature RISC complex then guides the recognition and binding of the target mRNA, leading to post-transcriptional gene silencing [117]. The miRNA molecules possess a key domain known as the seed sequence, which comprises the 2nd to 7th bases. This seed sequence plays a critical role in recognizing and binding to target mRNA through Watson–Crick base pairing [118]. In *M. japonicus*, the pri-miRNAs transcribed from the genome undergo processing by the enzyme Drosha, resulting in the generation of pre-miRNAs. Subsequently, these pre-miRNAs undergo additional processing into mature miRNAs within the cytoplasm, facilitated by the enzyme Dicer1 [119,120]. The mature miRNAs associate with the Ago1 protein in shrimp, forming the RISC, which facilitates post-transcriptional gene silencing [63,119,120,121].

Based on the discoveries made in shrimp, it can be inferred that the pathways involving miRNA-mediated and siRNA-mediated mechanisms are largely separate in these organisms. The production of siRNA or miRNA duplexes is carried out by different enzymes, namely Dicer 1 or Dicer 2, and these duplexes are then sorted into functionally distinct RISC that contain either Ago1 or Ago2 proteins. These findings align with similar observations made in fruit flies [122] but differ from what has been observed in mammals [123].

Recent investigations have demonstrated increasing evidence highlighting the crucial roles of miRNAs in the regulation of antiviral responses in aquatic organisms [63]. Studies have reported that shrimp miRNAs can exert antiviral effects by targeting specific genes, leading to the promotion of cellular phagocytosis and apoptosis, thus suppressing virus infection [63,124,125]. For instance, in WSSV infection, the viral miRNA WSSV-miR-N24 directly acts on the shrimp *caspase 8* gene, inhibiting apoptosis and consequently resulting in an increased number of WSSV copies in *M. japonicus* shrimp [124]. Furthermore, shrimp miRNAs have been found to activate multiple immune pathways simultaneously, leading to the suppression of virus infection [125]. An example is miR-12 in shrimp (*M. japonicus*), which targets shrimp genes *BI-1* (transmembrane BAX inhibitor motif containing 6) and *PTEN* (phosphatase and tensin homolog), as well as the viral gene *wsv024*, thereby triggering phagocytosis, apoptosis, and antiviral immunity [125]. Additionally, shrimp miRNAs have been implicated in the interplay between cellular autophagy and virus infection [126].

## 9. Evaluation of RNAi as a Remedy to Combat Viral Infection

RNAi offers a suitable alternative strategy by degrading the mRNA responsible for crucial viral proteins, effectively preventing the production of functional viral particles [127,128]. The efficacy of RNAi has been demonstrated in combatting various viral infections [129,130]. Numerous research investigations have employed sequence-specific dsRNA to impede the replication of viruses such as TSV, IHHNV, YHV, WSSV, LSNV, and GAV in shrimp [71,87,88,90,103]. These studies involve the targeted suppression of particular genes such as *Vp28* [67], *Vp37* [65], *rr2* [70], *V9* [97], *rab7* [80], a caspase-3 protein [131], and others (Table 2).

The application of dsRNA to a primary culture of black tiger shrimp lymphoid cells (Oka cells) demonstrated defense against YHV infection [72]. Similarly, the introduction of dsRNA through injection conferred immunity to WSSV and TSV in Pacific white shrimp, *L. vannamei* [132]. It was possible to achieve a systemic and dose-dependent reduction in YHV infection in *P. monodon* by specifically administering certain dsRNA molecules [71,73,132]. Research conducted using dsRNA targeting a potential protease found in TSV demonstrated that dsRNA with a specific sequence effectively suppressed TSV replication (resulting in an 11% mortality rate) in orally infected shrimp. In contrast, the control group experienced a complete mortality rate of 100% by the fifth-day post-infection [71]. Specific dsRNA has effectively hindered the replication of IHHNV or HPV. In an in vivo trial involving the intramuscular administration of dsRNA targeting YHV protease, the treated shrimp exhibited no mortality (0%) after 10 days following the challenge. In contrast, the control group experienced over 90% mortality [73]. Prophylactic administration of in vitro transcribed long dsRNA corresponding to viral genes *vp28, vp281*, and protein kinase in *Fenneropenaeus chinensis* (Osbeck) resulted in survival rates of 100%, 53%, and 93%, respectively, when administered up to 3 days before viral challenge [133]. Longer dsRNA molecules have the potential to generate a more diverse range of effective siRNAs incorporated into the RISC compared to shorter counterparts [72].

The application of YHV-specific dsRNA to shrimps infected with the virus within 12 h of infection completely halted viral replication and averted shrimp mortality [92]. This finding implies that besides its preventive role, RNAi could also be employed as a curative approach. In a different investigation, the complete elimination of WSSV in juvenile *P. japonicus* was achieved through a series of three consecutive injections of vp28-siRNA administered at 0, 24, and 48 h after the challenge [64]. Numerous investigations have been conducted concerning dsRNA targeting WSSV, primarily due to its significant threat in shrimp farming. Varied levels of success have been attained in suppressing WSSV replication through the utilization of sequence-specific dsRNA directed at diverse genes responsible for both structural and non-structural proteins. Despite the majority of these studies testing RNAi’s ability to hinder diseases or viral activity being conducted on a small scale, the positive results strongly suggest RNAi’s substantial potential for improving shrimp survival in aquaculture.

## 10. Delivery Strategies of RNAi Molecules

The method of delivering therapeutic molecules in RNAi-based therapy is a crucial concern [134]. Gaining insights into gene functionality and managing diseases in diverse aquatic organisms, it is essential to optimize effective protocols for delivering RNA molecules into cells or organisms. RNAi holds great potential for treating a wide range of diseases. However, the successful translation of RNAi from the laboratory to real-world applications faces challenges in delivering RNA molecules to specific cells of therapeutic interest within the organism. Various barriers exist, both inside and outside cells, that require careful design of delivery strategies. A variety of delivery techniques have been developed to effectively transport siRNA molecules, both in laboratory settings and in living organisms. These methods include electroporation, microinjection, oral administration, polymer-based systems, protein-based systems, and lipid nanoparticles [134] (Figure 2). The approach chosen will depend on the objectives of the study, the cell types that will be targeted, and/or the accessibility of the target [135]. There are benefits and drawbacks to each delivery strategy.

In 1999, the initial trial to introduce foreign DNA into *P. monodon* shrimps was carried out using the electroporation delivery technique [48]. Subsequently, electroporation has been employed multiple times for introducing foreign DNA into other organisms such as *Litopenaeus schmitti*, (Burkenroad) and *Artemia* [136,137]. The electroporation technique has also been utilized to introduce siRNA molecules into embryos of a model shrimp species (*Artemia sinica*) to downregulate the *As-sumo-1* gene [138]. These studies provide support for the efficacy of the electroporation technique in delivering nucleic acids into fish and shellfish embryos during early developmental stages. Furthermore, electroporation enables the delivery of nucleic acids to a large number of zygotes or embryos simultaneously [134].

The microinjection technique has been extensively employed for introducing nucleic acids, including dsRNA, into various fish and shellfish species at different developmental stages. Several studies have utilized microinjection to deliver siRNA to aquatic animals, yielding varying degrees of success. Examples of these studies include the delivery of siRNA to *Daphnia magna* (Straus) [139], *Macrobrachium rosenbergii* (De Man) [140], *P. vannamei* [141], and *P. monodon* [142]. The injection approach is employed when a precise amount is needed to be administered or in cells that require extra safety, such as zygotes [135].

Various transfection reagents have been utilized to introduce siRNA into diverse cell lines. Lipofectamine 2000 and Oligofectamine (Invitrogen) are commonly employed for siRNA delivery. Currently, no chemical transfection method for delivering dsRNA to shrimp has been reported. However, in an experiment involving DNA transformation in *L. vannamei*, successful results were achieved by microinjection, electroporation, and the use of the jetPEI transfection reagent method. This suggests that jetPEI has the potential to introduce dsRNA into shrimp embryos, at least in the case of this specific shrimp species [143,144].

Vector-based delivery methods encompass approaches that utilize plasmids and viruses. Viral vectors have been acknowledged as an effective delivery strategy for RNAi technology. However, their usage in aquaculture is restricted due to concerns about triggering detrimental immune responses and the potential risk of integrating into the host genome [145,146].

The administration of RNAi therapeutic molecules through oral delivery has shown success in various arthropod species. This can be achieved by delivering the molecules in a naked form, conjugated with a polymer, or using bacteria that carry specific dsRNA/siRNA. Additionally, incorporating the molecules into the feed of fish or shellfish has also been effective [147,148,149]. The utilization of nanoparticulate RNAi to specifically target the WSSV *vp28* gene resulted in significant antiviral efficacy. The study successfully suppressed the gene’s activity in both healthy and diseased models, providing substantial protection against viral challenges [150]. The most effective delivery of RNAi molecules has been achieved by the oral intake of dsRNA-enriched bacteria and viable brine shrimp (*Artemia*) zygotes [135]. However, the oral route stands out as the most encouraging approach for delivering RNAi in aquatic environments [151].

## 11. Challenges in RNAi Therapy

RNAi has gained recognition as a potent technique for modulating gene function and is seen as a promising approach for disease pathogen control in aquaculture. Despite the enthusiasm surrounding this impressive biological mechanism for precise gene regulation, several challenges and considerations need to be addressed before RNAi therapy can be practically implemented in shrimp aquaculture. These challenges include the risk of off-target effects, the activation of innate immunity, and notably, the successful administration of RNAi agents in vivo.

Ensuring the specificity of RNAi molecules is crucial to avoid unintended silencing of non-target genes. Off-target effects, where unintended genes are suppressed, can lead to undesirable consequences and potential side effects [152]. The theoretical specificity of RNAi is not fully realized, and off-target effects represent a significant challenge in the field of siRNA-based therapeutics. The comprehensive impact of individual siRNAs on the entire genome is largely unknown and difficult to predict. Extensive research has revealed that many experimentally validated off-target effects are associated with a 6 to 7 nucleotide match in the “seed” region of the siRNA [153,154,155]. Additionally, studies have shown that even an 11nucleotide match between the siRNA and unintended targets can lead to off-target gene silencing [156]. In vitro studies conducted on cell lines have indicated that off-target effects can adversely impact cellular viability and result in a toxic cellular phenotype [157]. Creating optimal siRNA duplexes or shRNA designs is a key approach for reducing the occurrence of unintended effects. Various factors have been identified in the literature that can impact the specificity of siRNAs/shRNA, including the selected target region [158], size [159], starting nucleotide [160], GC content [160], thermodynamic properties [161], and the presence of internal repeats or palindromes [162]. Consequently, several computational design tools have been created to systematically and precisely assess the off-target impacts of RNAi between siRNA sequences and target genes across the entire transcriptome [163,164,165].

The effective administration of RNAi molecules to particular cells or tissues remains a significant hurdle to developing a safe and successful in vivo RNAi therapy. siRNAs/shRNAs are prone to quick excretion, nonspecific tissue distribution, poor cellular absorption, low stability, and ineffective release within cells [166]. To enhance their stability in intracellular and extracellular environments, chemical modifications such as altering the backbone, substituting nucleotides with analogs, and incorporating conjugates can be introduced into the RNA oligos [167]. However, naked siRNAs, for their negative charge and size, generally cannot penetrate the cell membrane and reach the cytoplasm of the target cell, which is essential for efficient gene silencing [168]. Therefore, it is vital to possess suitable delivery techniques that enhance the concentration of siRNAs within cells and assist in their liberation from endosomes into the cytoplasm [169]. Improper selection of a delivery vector can diminish gene-silencing activity, increase unintended off-target effects, and lead to toxicity [170]. Overcoming the barriers of cellular and tissue-specific delivery is crucial for effective RNAi therapy.

Another crucial practical concern that must be considered is the potential for certain viruses to evade RNAi-mediated suppression. This evasion can occur through mutations in the targeted region and the presence of viral suppressors [171]. To counteract the emergence of resistant viruses, one strategy is to simultaneously target multiple viral sequences using a pool of siRNAs [172]. Another commonly employed alternative is the administration of long hairpin RNAs (lhRNA), which produce several siRNAs from one precursor without triggering an interferon (IFN) response [173]. However, it is important to note that the over-expression of multiple siRNAs may saturate the endogenous RNAi pathway, leading to undesired effects [174].

Additionally, RNAi molecules and other nucleic acids have the potential to trigger immune responses by being perceived as viral infections, which activate the interferon system [135]. The immunogenicity of RNAi molecules, along with the delivery systems used, can result in undesired immune reactions or diminished therapeutic effectiveness. It is crucial to effectively handle the immune response and minimize any immune-related adverse effects in order to ensure the success of RNAi therapy. Addressing these challenges through ongoing research and technological advancements will contribute to the successful implementation of RNAi therapy and its potential as a targeted therapeutic approach for various diseases.

## 12. Conclusions

RNAi offers a powerful tool to combat viral, bacterial, and parasitic infections that pose significant threats to shrimp populations. By targeting specific genes involved in the pathogen’s life cycle or the shrimp’s immune response, RNAi can effectively silence the expression of these genes, inhibiting the growth and replication of pathogens. This approach could help mitigate disease outbreaks, reduce mortality rates, and improve the overall health of shrimp stocks. Additionally, RNAi can be used to enhance the shrimp’s innate immune system, boosting their resistance against various pathogens.

The advancement of RNAi technology, from its initial discovery to potential clinical applications, has been remarkable. RNAi has recently emerged as a powerful tool for gene inhibition and shows promise in managing viral diseases in shrimp. The excitement and anticipation surrounding RNAi are well-founded; however, there are several challenges and considerations that need to be addressed for the practical application of this modern technology in aquaculture. These include the careful design of RNA constructs, optimizing the dosage, selecting an effective delivery strategy, implementing chemical modifications to enhance stability, and improving cellular uptake.

The findings and insights discussed in this review offer great promise for RNAi- based therapy. With the increasing occurrence of viral outbreaks among shrimp populations, there is an urgent requirement for enhanced therapeutic measures to effectively manage and control these diseases. If the challenges mentioned earlier can be addressed in a logical and systematic manner, RNAi-based therapies hold significant potential for combating these viral pathogens in shrimp aquaculture compared to conventional approaches, particularly in terms of specificity, targeted action and reduced environmental impact. Despite the obstacles that lie ahead, RNAi remains the most promising avenue for developing powerful and innovative therapeutic approaches against shrimp viral diseases. However, further research and development efforts are required to overcome the existing challenges and bring this game-changing remedy into practical application.

## Figures and Tables

**Figure 1 viruses-15-02050-f001:**
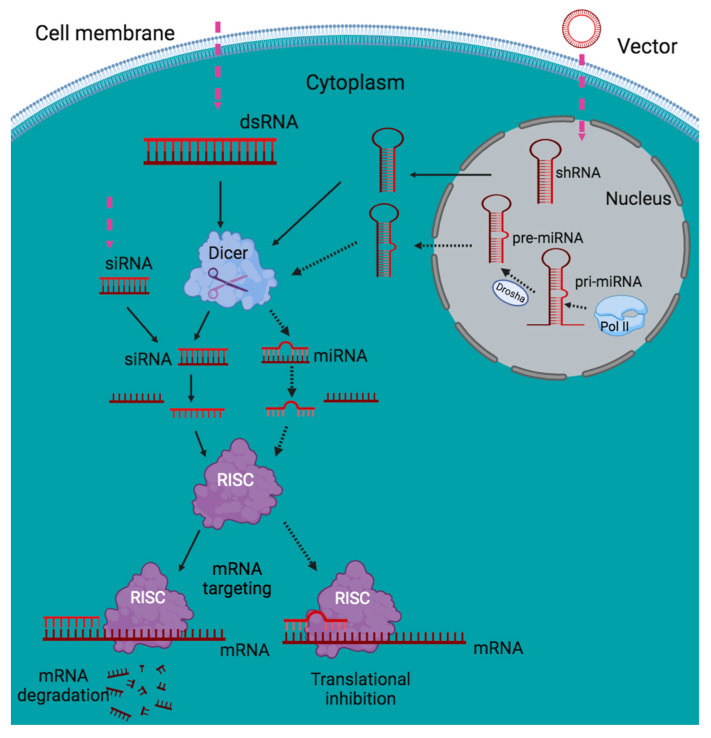
Schematic illustration of RNAi pathway (endogenous and exogenous). The siRNA and miRNA pathways are denoted by solid and dotted arrows, respectively. In the siRNA pathway, Dicer cleaves dsRNA to produce siRNAs in the cytoplasm. These siRNAs are integrated into the RISC complex and then unwound into two single-stranded RNAs known as the passenger and guide strand. The guide strand stays inside the RISC to control the sequence-specific decay of complementary mRNA while the exonucleases break down the passenger strand. On the other hand, the miRNA pathway starts with endogenously encoded pri-miRNAs, which are then transcribed by RNA polymerase II and trimmed by Drosha to produce pre-miRNAs. These precursors are subsequently transported to the cytosol where Dicer cleaves them to create miRNAs. Similar to the siRNA pathway, the miRNA duplex unwinds and one of the strands, the so-called mature miRNA, assembles into RISC, causing either mRNA breakage or translation inhibition, depending on the degree of similarity between the miRNA and the mRNA target. Exogenous RNAi triggers (pink dashed arrows), such as vector-based shRNAs and synthetic dsRNAs, siRNAs, and miRNAs, may also induce RNAi.

**Figure 2 viruses-15-02050-f002:**
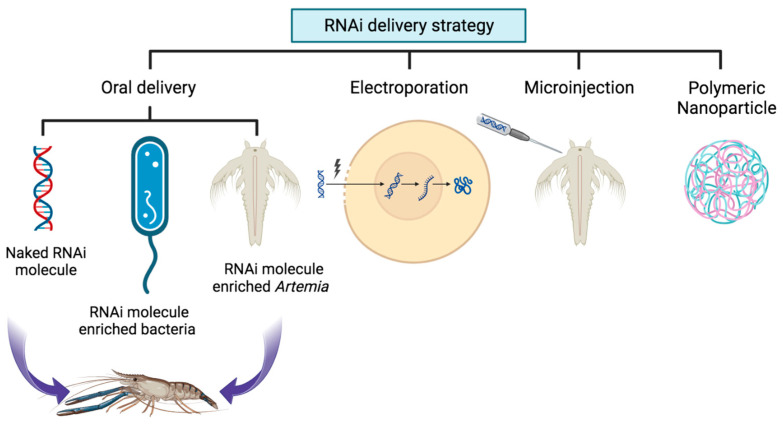
Potential delivery methods of RNAi molecule in shrimp to combat viral diseases.

**Table 1 viruses-15-02050-t001:** Different viral pathogens of shrimp.

Virus	Abbreviation	Family	Genome Type	Year of Emergence	Country of First Appearance	Reference
Infectious hypodermal and hematopoeitic necrosis virus	IHHNV	Parvoviridae	ssDNA	1981	Hawaii, USA	[20]
White spot syndrome virus	WSSV	Nimaviridae	dsDNA	1992	Taiwan	[21]
Hepatopancreatic parvovirus	HPV	Parvoviridae	ssDNA	1984	China	[22]
Spawner-isolated mortality virus	SMV	Parvoviridae	ssDNA	1993	Australia	[23]
Lymphoidal parvo-like virus	LPV	Parvoviridae	DNA	1991	Australia	[23]
Baculovirus penaei	BP	Baculoviridae	dsDNA	Late 1990s	Mexico	[24]
Monodon baculovirus	MBV	Baculoviridae	dsDNA	1977	Taiwan	[4,25]
Shrimp hemocyte iridescent virus	SHIV	Iridoviridae	DNA	2014	China	[26]
Taura syndrome virus	TSV	Picornaviridae	(+) ssRNA	1992	Ecuador	[4]
Infectious myonecrosis virus	IMNV	Totiviridae	(+) ssRNA	2002	Brazil	[9]
Covert mortality nodavirus	CMNV	Nodaviridae	RNA	2009	China	[27]
Penaeus vannamei nodavirus	PvNV	Nodaviridae	RNA	2004	Belize	[28]
Yellow head virus	YHV	Roniviridae	(+) ssRNA	1990	Thailand	[4]
Gill associated virus	GAV	Roniviridae	(+) ssRNA	1996	Australia	[29,30]
Laem Singh virus	LSNV	Leuteovirus-like	(+) ssRNA	2003	Thailand	[31]

## Data Availability

Data sharing not applicable.

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
