# Peer review of "RNAi-Based Therapy: Combating Shrimp Viral Diseases"

_viruses, 2023, doi:10.3390/v15102050_

Round 1

Reviewer 1 Report

In the review ‘RNAi-Based Therapy: Combating Shrimp Viral DiseasesMd. Shahanoor Alam and coauthors provides an overview of the potential of RNA interference as a therapeutic approach for controlling viral infections in shrimp. They highlight the advantages of RNAi-based therapy, including its specificity and potential for reducing the use of antibiotics. They also discuss the challenges and limitations of this approach, such as delivery methods and off-target effects. Overall, the review article provides valuable insights into the use of RNAi-based therapy for shrimp viral diseases. The review article is structured, and easy to follow. I suggest the following minor changes:

Please include a reference column in table 1.

Check the scientific name through the whole manuscript. When the scientific name appears first time in manuscript, it should be full written (both Genus and species), and then use the abbreviated binomial form for the following times. For example, line 107: P. monodon (should be full name, if appearing for the first time).

Line 270: Marsupenaeus japonicus should be abbreviated binomial form.

Not clear what is N.B. in figure 1 legend?

Author Response

Response to Reviewer 1 Comments

Comment 1: Please include a reference column in table 1.

Response 1: We greatly appreciate the reviewer’s efforts to carefully review the paper. This point is particularly helpful to improve our manuscript. As suggested by the reviewer, we have included a reference column in table 1. Please see the page 4.

Comment 2: Check the scientific name through the whole manuscript. When the scientific name appears first time in manuscript, it should be full written (both Genus and species), and then use the abbreviated binomial form for the following times. For example, line 107: P. monodon (should be full name, if appearing for the first time).

Line 270: Marsupenaeus japonicus should be abbreviated binomial form.

Response 2: Thanks for the comments. We have checked the scientific names thoroughly the whole manuscript. According to your suggestions we have corrected the scientific names in the manuscript. In line 107 and 270, we have corrected the scientific names following your suggestions.

Comment 3: Not clear what is N.B. in figure 1 legend?

Response 3: Basically, N.B. is an abbreviation for the Latin phrase "nota bene", meaning “note well.” It is used to emphasize an important point.

Reviewer 2 Report

This review is very simplistic in its approach and informs very short and basic about different virus on shrimp ( why not at least crustaceans), immunity in shrimp (why not al least in crustaceans) and some other additional items on shrimp. The review title is all about RNAi based methods for combating virus infection in shrimp and this should be the main and only subject of this mini-review and not more or less school book knowledge about shrimp and shrimp diseases and immunity. Therefore this reviewer feels that the review should be focussing only on what the title implies and if acceptable for the Editor is has to be extensively shortened and focussed on the RNAi based technology used in shrimp(crustaceans) up to date.  

Needs editing

Author Response

Response to Reviewer 2 Comments

Comments:This review is very simplistic in its approach and informs very short and basic about different virus on shrimp (why not at least crustaceans), immunity in shrimp (why not at least in crustaceans) and some other additional items on shrimp. The review title is all about RNAi based methods for combating virus infection in shrimp and this should be the main and only subject of this mini-review and not more or less school book knowledge about shrimp and shrimp diseases and immunity. Therefore, this reviewer feels that the review should be focussing only on what the title implies and if acceptable for the Editor is has to be extensively shortened and focussed on the RNAi based technology used in shrimp(crustaceans) up to date.  

 Responses: We greatly appreciate your thoughtful feedback. Your points are well-founded, and we fully acknowledge the significance of closely aligning the content of our review with its intended title and subject matter. The primary objective of our review was, indeed, to provide a comprehensive exploration of the potential applications of RNAi-based methods for combating viral infections in shrimp. Our review primarily focuses on shrimp rather than other crustaceans as a whole because shrimp is a key commercial product within the industry.

We agree that it should remain focused on this specific topic and avoid delving into broader discussions about shrimp biology, diseases, or immunity unless directly relevant to RNAi methods. But it is worth noting that RNAi is an integral component of the innate immunity defense mechanism in shrimp. Consequently, aspects related to shrimp biology, existing shrimp viral diseases, and immunity are undeniably relevant and should be included in our review, provided they are somehow linked to RNAi. Furthermore, it's important to note that while focusing on shrimp RNAi combating viral diseases, we also discussed and compared it to other current therapeutic strategies to minimize the impacts of viral diseases in shrimp aquaculture. To establish the competence of shrimp RNAi, it is essential to compare it with current therapeutic strategies in the field. This comparison allows for a comprehensive evaluation of the effectiveness and potential advantages of RNAi-based method in managing viral infections in shrimp. Indeed, comparing shrimp RNAi with current therapeutic strategies can add depth to the review and make it longer than originally intended.

We genuinely value your feedback, as it plays a vital role in enhancing both the quality and alignment of our content with its stated subject matter. Your commitment to elevating the quality of our work is greatly appreciated. We have checked and revised the English language for better readable according to your suggestions.

Reviewer 3 Report

The review provides an informative and comprehensive overview of the promising RNAi-based therapy for combating shrimp viral diseases. I only have the following minor comments for improvement:

-Species authorities should be added upon first mentioning of a species name throughout the manuscript, starting with Marsupenaeus japonicus. Depending on the journal, this might be done without indicating the year of the original description (usually the reference is not cited in the list of references then) or with indication of the year of the original description, with the reference to be included in or excluded from the list of references.

-The last paragraph of the Introduction section lacks citations.

-Paragraph 2.1. needs some rearrangement as there is no clear red thread, e.g. lines 85 and lines 91-92 refer to the time to death but other aspects are mentioned in between.

-Line 248: Which species is meant with “P. clarkia”? Procambarus clarkii?

-Paragraph 4.5. This paragraph lacks citations.

-Table 2: “P.monodon,Mjaponicus” needs to be corrected.

-Caption to Figure 1: This caption is too long and the explanation of the pathway should be integrated in the manuscript body instead.

-12. Conclusion (maybe better write Conclusions instead): The conclusion should not only deal with RNAi but also briefly compare it with other approaches, such as elevated temperature.

English language is mostly fine, only some minor corrections and rearrangements are required.

Author Response

Response to Reviewer 3 Comments

Comment 1: Species authorities should be added upon first mentioning of a species name throughout the manuscript, starting with Marsupenaeus japonicus. Depending on the journal, this might be done without indicating the year of the original description (usually the reference is not cited in the list of references then) or with indication of the year of the original description, with the reference to be included in or excluded from the list of references.

Response 1: We greatly appreciate the reviewer’s efforts to carefully review the manuscript. We have added the species authorities upon first mentioning of a species name throughout the manuscript.

Comment 2: The last paragraph of the Introduction section lacks citations.

 Response 2: This point is particularly helpful to improve our manuscript. As suggested by the reviewer, we have cited the mentioned paragraph adequately. Please see the page 2.

Comment 3: Paragraph 2.1. needs some rearrangement as there is no clear red thread, e.g. lines 85 and lines 91-92 refer to the time to death but other aspects are mentioned in between.

Response 3: We have rearranged the Paragraph 2.1 following the reviewers’ suggestions. Please see the page 2, lines 87-95.

Comment 4: Line 248: Which species is meant with “P. clarkia”? Procambarus clarkii?

Response 4: P. clarkia was mistakenly written. It would be basically Procambarus clarkii. We have corrected the spelling of the species in the manuscript. Please see the page 6, line 253.

Comment 5: Paragraph 4.5. This paragraph lacks citations.

Response 5: We have cited the mentioned paragraph adequately. Please see the page 6.

 Comment 6: Table 2: “P.monodon,Mjaponicus” needs to be corrected.

Response 6: We have corrected “P.monodon,Mjaponicus” in the manuscript. It would be as P. monodon, M. japonicus. Please see the page 8.

 Comment 7: Caption to Figure 1: This caption is too long and the explanation of the pathway should be integrated in the manuscript body instead.

Response 7: Thank you for your valuable comment. We agree with the reviewer that Figure 1 caption is relatively longer. Although RNAi pathway has been clearly described in the manuscript, the illustration demands a very short explanation as we used here different signs (solid and dotted arrows, colors etc.) for better understanding of the reader. However, we have shortened the Caption of Figure 1. Please see the page 9.

Comment 8: 12. Conclusion (maybe better write Conclusions instead): The conclusion should not only deal with RNAi but also briefly compare it with other approaches, such as elevated temperature.

Response 8: Thanks for this comment and suggestion. We have corrected “Conclusion” as “Conclusions”. As reviewer’s suggestions, we have also tried to compare RNAi therapy with other conventional shrimp viral diseases management approaches in terms of efficacy. Please see the page 15.

Round 2

Reviewer 2 Report

The authors have only moderately edited and revised the manuscript and this reviewer therefore is unable to recommend it for inclusion in this or any other journal. 

The English is OK

Author Response

We deeply appreciate your valuable input. Your observations are sound, and we wholeheartedly recognize the importance of aligning our review's content closely with its intended title and subject matter. According to your suggestions we have extensively shortened and focused on the RNAi based technology used in shrimp. Consequently, we have meticulously edited and refined the manuscript.

We sincerely value your dedication to enhancing the quality of our work.
